

# Inter-set rest configuration effect on acute physiological and performance-related responses to a resistance training session in terrestrial *vs* simulated hypoxia

Cristina Benavente[1], Belén Feriche[1], Guillermo Olcina[2], Brad J. Schoenfeld[3], Alba Camacho-Cardenosa[2], Filipa Almeida[1], Ismael Martínez-Guardado[4], Rafael Timon[2] and Paulino Padial[1]

[1] Department of Physical Education and Sport, Faculty of Sport Sciences, University of Granada, Granada, Spain
[2] Faculty of Sport Sciences, University of Extremadura, Cáceres, Spain
[3] Department of Health Sciences, CUNY Lehman College, New York, United States of America
[4] Faculty of Education, BRABE Group, Department of Psychology. Faculty of Life and Nature Sciences, University of Nebrija, Madrid, Spain

Corresponding author
Belén Feriche, mbelen@ugr.es

## ABSTRACT

**Background**. Metabolic stress is considered a key factor in the activation of hypertrophy mechanisms which seems to be potentiated under hypoxic conditions. This study aimed to analyze the combined effect of the type of acute hypoxia (terrestrial *vs* simulated) and of the inter-set rest configuration (60 *vs* 120 s) during a hypertrophic resistance training ($R_T$) session on physiological, perceptual and muscle performance markers.

**Methods**. Sixteen active men were randomized into two groups based on the type of hypoxia (hypobaric hypoxia, HH: 2,320 m asl; *vs* normobaric hypoxia, NH: $FiO_2$ of 15.9%). Each participant completed in a randomly counterbalanced order the same $R_T$ session in four separated occasions: two under normoxia and two under the corresponding hypoxia condition at each prescribed inter-set rest period. Volume-load (load × set × repetition) was calculated for each training session. Muscle oxygenation ($SmO_2$) of the vastus lateralis was quantified during the back squat exercise. Heart rate (HR) was monitored during training and over the ensuing 30-min post-exercise period. Maximal blood lactate concentration (maxLac) and rating of perceived exertion (RPE) were determined after the exercise and at the end of the recovery period.

**Results**. Volume-load achieved was similar in all environmental conditions and inter-set rest period length did not appreciably affect it. Shorter inter-set rest periods displayed moderate increases in maxLac, HR and RPE responses in all conditions. Compared to HH, NH showed a moderate reduction in the inter-set rest-HR (ES > 0.80), maxLac (ES > 1.01) and $SmO_2$ (ES > 0.79) at both rest intervals.

**Conclusions**. Results suggest that the reduction in inter-set rest intervals from 120 s to 60 s provide a more potent perceptual, cardiovascular and metabolic stimulus in all environmental conditions, which could maximize hypertrophic adaptations in longer periods of training. The abrupt exposure to a reduced $FiO_2$ at NH seems to reduce the inter-set recovery capacity during a traditional hypertrophy $R_T$ session, at least during a single acute exposition. These results cannot be extrapolated to longer training periods.

## INTRODUCTION

The increase of muscle mass and strength *via* resistance training ($R_T$) is a primary goal for athletes, recreationally trained individuals, and populations interested in improving various health-related outcomes (*Schoenfeld, 2010*). The results of a training program may vary depending on the manipulation of several variables including training volume (sets × repetitions × load), inter-set rest period length, movement velocity, exercise selection, exercise sequence and training frequency (*Bird, Tarpenning & Marino, 2005*). Training volume and load are considered primary factors to maximize strength and hypertrophy (*Kraemer & Ratamess, 2004*), but other variables, such as rest intervals, also play an important role in both acute and chronic responses to $R_T$ programs (*De Salles et al., 2009*). Hypertrophy training is associated with the use of short (<60) to long (>90 s) inter-set rest-intervals (*Henselmans & Schoenfeld, 2014*). Both, short and long rest intervals, can be used to enhance strength and muscle growth: although mechanisms remain speculative, it has been hypothesized that short rest periods induce beneficial effects *via* increased metabolite accumulation while long intervals provide a greater capacity to maintain high training intensities and volume load (*Wernbom et al., 2007*; *Schoenfeld, 2010*).

Evidence suggests that $R_T$ performed under hypoxic conditions may produce an added benefit to strength and muscle mass development compared to an equivalent amount of training under normoxic conditions (*Nishimura et al., 2010*; *Manimmanakorn et al., 2013a*; *Manimmanakorn et al., 2013b*). This benefit is purportedly linked to the heightened accumulation of metabolic byproducts in hypoxia, such as blood lactate, protons ($H^+$), calcium and inorganic phosphorus, among others, derived from the increase in anaerobic metabolism to compensate the loss of oxygen ($O_2$) availability (*Kon et al., 2012*; *Schoenfeld, 2013*; *Kurobe et al., 2015*; *Scott, Slattery & Dascombe, 2015*). Metabolic stress has been proposed as a factor in the activation of muscle hypertrophy-related mechanisms (*i.e.,* activation of anabolic signaling routes) (*Schoenfeld, 2010*; *Schoenfeld, 2013*). Current evidence indicates that multiple sets of high-intensity $R_T$ lead to significant acute physiological responses (*Schoenfeld, 2013*; *Cintineo et al., 2018*) also mediated by inter-set rest configuration, both under conditions of normoxia (*De Salles et al., 2009*; *Henselmans & Schoenfeld, 2014*; *Grgic et al., 2018*) and hypoxia (*Lockhart et al., 2020*). In addition, it has been proposed that the accumulation of metabolites promotes the recruitment of additional high-threshold motor units (*Miller et al., 1996*; *Takarada et al., 2000*; *Debold, 2012*), increasing the total number of muscle fibers stimulated (*Scott, Slattery & Dascombe, 2015*).

In regard to hypoxic training, it is important to consider how the type of the hypoxia and its interaction with the manipulation of training variables might influence the $R_T$ response. Systemic hypoxia can be achieved by the ascent to high altitudes (hypobaric hypoxia (HH)) or by breathing $O_2$-depleted air (normobaric hypoxia (NH)). Current data suggest that the physiological response differs between both modalities of hypoxia due to

factors related to the barometric pressure and/or partial pressure of $O_2$ (*Millet & Debevec, 2020*). However, current literature does not sufficiently address the physiological effects of a $R_T$ period at terrestrial altitude and results obtained from NH studies are inconclusive. This is likely due to the methodological heterogeneity in exercise protocols and in the level of hypoxia used among studies (*Feriche et al., 2017*; *Ramos-Campo et al., 2018*).

It has been hypothesized that $R_T$ in hypoxia might only provide additional benefits when relatively short inter-set rest periods are used, while longer rest periods could mitigate any effects of hypoxia on the muscle environment (*Scott et al., 2015*). However, the availability of studies comparing the effect of recovery time between sets in hypoxia is scarce. From the results of research using different inter-set rest periods under hypoxic conditions, shorter inter-set rest intervals (<60 s) have been shown to be effective in muscle activation and development at both acute (*Kon et al., 2010*) and chronic NH conditions (*Nishimura et al., 2010*; *Kurobe et al., 2015*). Contrarily, inter-set rest periods longer than 90 s did not provide benefits on the muscle response after a single $R_T$ session (*Scott, Slattery & Dascombe, 2015*; *Scott et al., 2015*) or after a longitudinal training period at NH (*Kon et al., 2014*; *Ho et al., 2014*). Similar results were observed at acute moderate HH with 120 s of inter-set rest intervals (*Feriche et al., 2020*), although the effects of shorter recoveries at this type of hypoxia remain unknown. As in normoxia, higher inter-set recovery times in hypoxia may also favor intramuscular metabolite clearance, limiting the potential benefit of metabolic stress on its putative anabolic effects, which in turn may disfavor muscle hypertrophy (*Scott, Slattery & Dascombe, 2015*). The only previous study to examine the effects of different rest periods during $R_T$ under H conditions (*Lockhart et al., 2020*) used a single-joint exercise, thus limiting the ability to draw strong inferences with regard to more complex or metabolically demanding protocols as to whether short rest intervals combined with different types of hypoxia may enhance muscular adaptations.

Considering the putative role that metabolic stress plays in the hypertrophic response to resistance exercise, the duration of inter-set rest may be an important consideration in exercise program design, particularly under hypoxic conditions. The aim of this study was to compare the effect of different types of acute hypoxia (HH *vs* NH) combined with different inter-set rest configurations (60 s *vs* 120 s) during a traditional hypertrophy-oriented $R_T$ session on perceptual, physiological and muscle performance markers. The results will help to determine the influence of the inter-set rest configuration on acute stress markers, which potentially could provide insight into strategies for optimizing strength and muscle mass gains over longer training periods. We hypothesized that short rest periods would produce higher perceptual, cardiovascular and blood lactate changes, and its combination with terrestrial hypoxia would maximize this response.

## MATERIALS & METHODS

### Experimental approach to the problem

Our research design allowed for comparisons of muscle performance markers to a hypertrophy training session between environmental conditions (HH *vs* NH) and exercise inter-set rest configuration (60 s *vs* 120 s) while controlling for other variables. A repeated

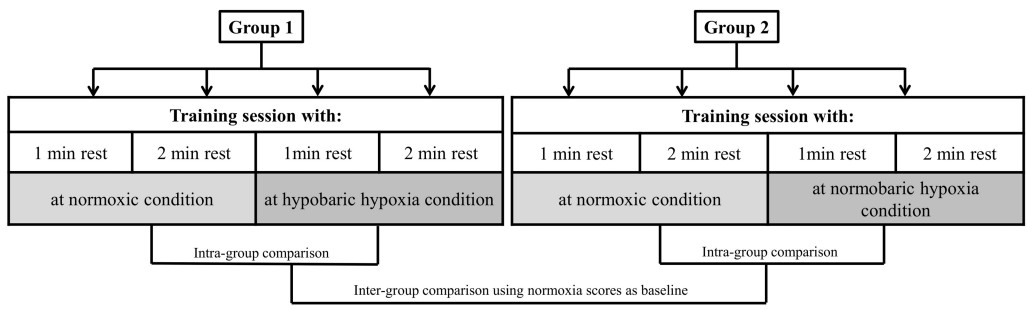

**Figure 1  Study design.**

measures model was applied in two independent groups (G1 and G2), one for each type of hypoxia. All participants performed a standard hypertrophic $R_T$ session on four different days, counterbalancing the order in terms of environmental condition and type of inter-set rest. Each session was separated by a rest period of 72 h. Thus, participants in G1 performed each of the two inter-set rest types of $R_T$ sessions at normoxia (N) and at terrestrial hypoxia (HH: 2,320 m asl; ~570 mmHg). Participants in G1 travelled by car to the HH center (32 km), began the training session ~30 min after arrival to altitude and then returned to normoxia after completing the session. Participants in G2 performed the same routines as G1 under equivalent simulated normobaric hypoxia (NH: <700 m asl; inspired fraction of oxygen [$FiO_2$] = 15.9%). The study design is illustrated in Fig. 1.

One week before the first $R_T$ session, subjects engaged in a preparatory session to determine their training load (70% of 1RM) for each exercise. This load was the average between two attempts with different loads separated by 15 min. Two days before the beginning of the study, participants attended the laboratory for baseline anthropometric measures (height: Seca 202; Seca Ltd., Hamburg, Germany) and body mass (Tanita BC 418 segmental; Tokyo, Japan)). Preliminary assessments were performed under normoxic conditions and participants were instructed to abstain from physical activity and alcohol intake, and to maintain their customary sleep and diet habits for 48 h before evaluations. To ensure standardized nutritional intake for performance during the $R_T$ sessions, participants fasted after midnight the evening prior to a training session and were provided with a standardized breakfast (730 kcal) and a protein bar (350 kcal) at 2 h and at 40 min prior to the start of the warm-up, respectively. Exercise was conducted in the morning at the same time of day for all participants under the conditions of ~22 °C and ~60% humidity for the N and NH conditions, or ~22 °C and ~28% humidity for the HH condition. The hypoxic environmental condition was assessed by the arterial oxygen saturation ($SpO_2$) measured before the start of the warm-up.

## Participants
Sixteen active, resistance-trained men (G1 [$n = 9$]; age: 23.6 ± 3.2 years; height: 177.2 ± 5.7 cm; body mass: 73.9 ± 5.3 kg and G2 [$n = 7$]; age: 26.0 ± 3.0 years; height: 174.0 ±

5.0 cm; body mass: $73.9 \pm 7.8$ kg) volunteered to participate in the study. Subjects had no self-reported health or muscular disorders and were not exposed to more than 3-4 consecutive days of altitudes above 1,500 m asl for at least two months before the study. Participants lived at a low altitude to ensure that responses were specific to acute hypoxia exposure. All subjects had been consistently lifting weights for at least 12 months prior to the onset of the study. Before the study, participants were provided with information detailing the purpose and requirements of the research protocol and provided signed informed consent. This study was approved by the Andalusian Government Research Ethics Committee (Ethical Application Ref: # 1540-n-18) and conducted in accordance with the Helsinki Declaration.

## Procedures
### Hypertrophic resistance training session
The $R_T$ session included six exercises that targeted movement patterns involving major muscle groups of the body in the following order: back squat, machine leg press, seated cable row, wide grip lat pulldown, bench press and barbell military press. Before the training sessions, participants undertook a standard warm-up protocol consisting of 15 min of low intensity aerobic exercise and stretching exercises, and a specific warm-up in which they performed two sets of 10 repetitions (the first with 20 kg and the second at 50% 1RM estimated from the preliminary test, 120 s rest) of the back squat, seated cable row and bench press.

Each training session comprised three sets of 10 repetitions per exercise with a load of 70% of 1RM and 60 s or 120 s of inter-set and inter-exercise rest. Cadence of repetitions was carried out in a controlled fashion, with a concentric action of approximately 1 s and an eccentric action of approximately 2 s as determined by the supervising researcher. The load was reduced by 5% as needed in those cases that participants reached volitional failure before achieving the target repetition range (8–10 repetitions) with respect to the previous set (*i.e.,* in the 2nd or 3rd set). All routines were directly supervised by the research team to ensure they were properly performed.

Absolute training load by exercise (kg) and repetitions were monitored during each training session. Due to differences in training machine models between locations, only the barbell back squat and bench press were used for comparison. Total volume-load was calculated as the sum of the load lifted × the repetitions × set of each exercise (*Scott et al., 2014*). Before the warm-up of each session $SpO_2$ was measured in duplicate using a pulse oximeter (Wristox 3100; Nonin, Plymouth, MN, USA). Participants mean rest $SpO_2$ value equated to $98.4 \pm 0.9$ and $94.3 \pm 0.5\%$ for G1 (N and HH, respectively), and $98.5 \pm 0.5$ and $90.7 \pm 1.0\%$ for G2 (N and NH, respectively).

### Hypobaric-normobaric hypoxia conditions
G1 performed the hypoxic training sessions under terrestrial hypoxic conditions at the High-Performance Center of Sierra Nevada (2320 m asl., Spain). The normobaric hypoxia condition of G2 was carried out by connecting a facial mask to participants 5 min before the start of the warm-up that maintained breathing at a reduced $FiO_2$ (15.9%) during the hypoxic training sessions. $FiO_2$ during exercise was controlled using an electronic device

(HANDI+, Maxtec, Salt Lake City, Utah, USA). The $FiO_2$ level was calculated according to the guidelines provided by the hypoxic generator manufacturer to equate an altitude of 2320 m. The low oxygen air was produced by a hypoxic generator with a semi-permeable filtration membrane (nitrogen filter technique; CAT 310, Louisville, Colorado, USA).

### Training session monitoring

*Metabolic and cardiovascular responses.* Blood lactate concentration (Lac) was assessed before and immediately following the training session, at minutes 3, 5, 10 and 30 using a Lactate Pro 2 device (Arkray, Japan). Basic cardiovascular response was quantified from a heart rate (HR) cardiotachometer (Polar s610i; Polar Electro Oy, Kempele, Finlandia) during all training sessions and over the course of the immediate 30 min post-exercise period. The mean value of HR recorded was classified as working HR (work-HR), rest time between sets HR (rest-HR) and HR along the post-exercise recovery period ($HR_{30}$).

*Perceptual responses.* Sessional rating of perceived exertion was obtained *via* a Category Ratio-10 scale viewed by participants 30 min after completing the training session (RPE-30) (*Day et al., 2004*).

*Muscle oxygenation.* Muscle oxygen saturation ($SmO_2$) was measured by near-infrared spectroscopy (NIRS; Moxy, Fortiori Design, Minneapolis, Minnesota, USA) during the first exercise (back squat) of each training session. The Moxy device measures the total hemoglobin (Hb) present beneath the device, as well as calculates the percentage of Hb containing $O_2$ ($SmO_2$) (*Crum et al., 2017*). $SmO_2$ reflects the dynamic balance between $O_2$ supply and consumption calculated throughout the change in total tissue oxy (+myo) hemoglobin ($O_2Hb$) and deoxyhemo- (+myo-) globin (HHb) (*McManus, Collison & Cooper, 2018*). The sampling rate of the sensor was 2 Hz. $SmO_2$ values were expressed in % and calculated as follows by the device:

$$SmO_2(\%) = O_2Hb/[O_2Hb + HHb] \times 100.$$

During all testing, the system was connected to a personal computer *via* a software program (Seego: Realtrack Systems, Almería, Spain) that provided a graphic display of the data. The sensor was placed on the vastus lateralis of the participant's dominant leg, halfway between the greater trochanter and lateral epicondyle of the femur, before the warm-up. This position was marked with a semi-permanent pen on the skin to reproduce the exact location in subsequent tests. To avoid issues with movement during exercise, the device was fixed to the leg with tape and wrapped with a dark elastic bandage. Maximal and minimum values were recorded for each set of the exercise. The difference between maximal and minimum values was used to calculate the $SmO_2$ of the first ($SmO_2S_1$), second ($SmO_2S_2$) and third ($SmO_2S_3$) set. The mean of the three sets was calculated to express the total mean $SmO_2$ of the exercise ($SmO_2T$).

### Statistical analyses

Data are presented as mean $\pm$ standard deviation (SD). Normal distributions of the data were confirmed using a Shapiro-Wilk test. A linear mixed-effects model with inter-set

recovery (60 s *vs* 120 s), environmental condition (HH and NH), and their interaction was applied for analysis. Varied intercepts were permitted by treating subject as a random effect. This model was built for the physiological variables. To ascertain the eventual effect of training load on performance of 2 comparable exercises among conditions (back squat for the lower-limbs and bench press for the upper-limbs), normoxia baseline scores were included as a covariate of no interest (*Bates et al., 2015*). Also, the adjusted between-group difference was calculated as the estimated marginal mean of the difference between HH and NH groups (HH group–NH group) after adjusting for N baseline differences. To quantify the magnitude of the change, we calculated 90% confidence intervals (CIs) of the adjusted effect.

The standardized mean differences (*i.e.,* Cohen's d effect sizes) were calculated as the mean change (H-N or 120-60 s) divided by the pooled standard deviations of the change in all dependent variables or as the adjusted between-group difference divided by the pooled normoxia SD when comparing hypoxia types (*Cohen, 1988*). Threshold classifications were set as follows: >0.2 [small], >0.6 [moderate], >1.2 [large] and >2 [very large] (*Hopkins et al., 2009*).

Consistent with other research in applied sports science (*Almeida et al., 2021*), we used an estimation-based approach to drawing inferences from our data. Accordingly, we interpreted each effect and its precision continuously (*Gardner & Altman, 1986*) rather than relying on null hypothesis significance testing (*Amrhein, Greenland & McShane, 2019*). This follows current statistical recommendations to eschew dichotomous interpretations of results in favor of models that provide estimates of practical meaningfulness (*Wasserstein, Schirm & Lazar, 2019*). All analyses were performed using the software package SPSS (version 26.0, IBM SPSS Statistics for Windows; IBP Corp., Armonk, NY, USA).

# RESULTS

## Resistance training session

Table 1 displays the mean total volume-load accumulated during the 3 sets of the 2 free barbell exercises across conditions. The adjusted between-group effects showed no meaningful differences in volume-load between both types of hypoxia at each of the inter-set rest intervals in the 2 analyzed exercises (adjusted between-group effect from $-7.64$ to 51.75 kg [90% CIs from $-135$ to 238.53 kg] and from $-43.05$ to $-15.55$ kg [90% CIs from $-110.03$ to 51.21 kg], respectively for 60 and 120 s inter-set rest intervals). However, trivial to moderate increases in the total volume-load were achieved at longer inter-set rest periods in the bench press at HH (5.9%, ES = 0.35, $p = 0.027$).

## Cardiovascular, metabolic and perceptual responses

Heart rate, blood lactate and RPE-30 responses are presented in Table 2. The results showed moderately lower mean work and rest-HR values with 120 s inter-set rest periods at normoxia (ES: from 1.01 to 1.08) and both types of hypoxia (ES: from 0.58 to 0.92). A similar work-HR response was observed between HH and NH conditions. However, we detected a lower mean rest-HR in NH during both inter-set rest intervals than in HH

Benavente et al. (2022), *PeerJ*, DOI 10.7717/peerj.13469

**Table 1  Total volume-load during the three training sets in both groups.**

| | | Total volume-load (Kg) | | | | | | | |
| --- | --- | --- | --- | --- | --- | --- | --- | --- | --- |
| | | G1 | | | G2 | | | HH vs NH | |
| | | N | HH | N vs HH ES [CI 90%] *p-value* | N | NH | N vs NH ES [CI 90%] *p-value* | Adjusted differences between hypoxia types [CI 90%] | ES [CI 90%] *p-value* |
| Back squat (kg) | 60 s | 2114.4 ± 517.8 | 2123.3 ± 468.6 | **0.02 [−0.29; 0.33]** *0.904* | 2142.9 ± 240.5 | 2100.0 ± 245.0 | **−0.18 [−0.48; 0.13]** *0.594* | 51.75 [−135.03; 238.53] | **0.14 [−0.75; 1.02]** *0.629* |
| | 120 s | 2111.7 ± 522.4 | 2096.1 ± 520.9 | **−0.03 [−0.15; 0.08]** *0.877* | 2100.0 ± 245.0 | 2100.0 ± 245.0 | – | −15.556 [−82.32; 51.21] | **−0.04 [−0.92; 0.84]** *0.676* |
| | 60 vs 120 s ES [CI 90%] *p-value* | **0.01 [−0.04; 0.05]** *0.976* | **0.06 [−0.21; 0.32]** *0.740* | | **0.18 [−0.13; 0.48]** *0.688* | – | | | |
| Bench press (kg) | 60 s | 1628.3 ± 353.1 | 1600.0 ± 275.4 | **−0.09 [−0.28; 0.10]** *0.608* | 1529.3 ± 307.6 | 1522.9 ± 275.0 | **−0.02 [−0.12; 0.07]** *0.902* | −7.64 [−75.77; 60.49] | **−0.02 [−0.91; 0.86]** *0.844* |
| | 120 s | 1773.3 ± 382.3 | 1700.6 ± 300.4 | **−0.21 [−0.37; −0.05]** *0.053* | 1537,1 ± 288.4 | 1541.4 ± 296.8 | **0.02 [−0.08; 0.10]** *0.925* | −43.05 [−110.03; 23.93] | **−0.13 [−1.01; 0.76]** *0.277* |
| | 60 vs 120 s ES [CI 90%] *p-value* | **−0.40 [−0.66; −0.13]** *0.009* | **−0.35 [−0.54; −0.16]** *0.027* | – | **−0.03 [−0.09; 0.03]** *0.894* | **−0.07 [−0.21; 0.08]** *0.617* | – | | |

**Notes.**

G1, Group 1; G2, Group 2; N, normoxic condition; HH, hypobaric hypoxia condition; NH, normobaric hypoxia condition; 60 s/120 s:, inter-set rest of the session; ES, effect size (calculated as mean difference (H-N or 120-60 s) ÷ (pooled SD) in all dependent variables).

Adjusted between-group difference is the estimated marginal mean of the difference between HH and NH groups (HH group–NH group) after adjusting for N baseline differences.

CI 90%, 90% confidence interval.

Intra- and inter-group ES [CI 90%] are shown in bold.

Benavente et al. (2022), *PeerJ*, DOI 10.7717/peerj.13469

**Table 2** Mean physiological and perceptual measures recorded in both groups with different inter-set rest and conditions.

| | | G1 | | | G2 | | | HH *vs* NH | |
|---|---|---|---|---|---|---|---|---|---|
| | | N | HH | N *vs* HH ES [CI 90%] *p-value* | N | NH | N *vs* NH ES [CI 90%] *p-value* | Adjusted differences between hypoxia types [CI 90%] | ES [CI 90%] *p-value* |
| Work-HR (bpm) | 60 s | 150.7 ± 14.3 | 147.8 ± 18.5 | −0.18 [−0.46; 0.10] 0.711 | 143.9 ± 13.0 | 144.8 ± 12.8 | 0.06 [−0.54; 0.67] 0.908 | 3.09 [−10.69; 16.87] | 0.20 [−0.69; 1.08] 0.699 |
| | 120 s | 136.2 ± 17.3 | 136.4 ± 21.3 | 0.01 [−0.16; 0.18] 0.984 | 120.2 ± 22.6 | 136.0 ± 13.9 | 0.87 [−0.25; 1.99] 0.097 | 0.40 [−15.15; 15.96] | 0.02 [−0.86; 0.91] 0.964 |
| 60 *vs* 120 s | ES [CI 90%] *p-value* | 0.92 [0.43; 1.41] 0.082 | 0.58 [0.24; 0.91] 0.241 | | 1.34 [0.15; 2.53] 0.015 | 0.66 [0.14; 1.17] 0.244 | | | |
| Rest-HR (bpm) | 60 s | 155.9 ± 14.2 | 154.0 ± 17.0 | −0.12 [−0.38; 0.14] 0.806 | 139.6 ± 14.7 | 140.5 ± 15.5 | 0.06 [−0.60; 0.72] 0.916 | 13.56 [−0.85; 27.97] | 0.83 [−0.09; 1.75] 0.120 |
| | 120 s | 139.9 ± 21.0 | 141.2 ± 22.5 | 0.06 [−0.12; 0.24] 0.906 | 110.0 ± 25.8 | 125.1 ± 18.1 | 0.69 [−0.43; 1.81] 0.189 | 16.12 [−1.73; 33.98] | 0.80 [−0.12; 1.71] 0.134 |
| 60 *vs* 120 s | ES [CI 90%] *p-value* | 0.91 [0.47; 1.34] 0.093 | 0.65 [0.33; 0.97] 0.193 | | 1.47 [0.27; 2.66] 0.008 | 0.92 [0.26; 1.57] 0.114 | | | |
| HR30 (bpm) | 60 s | 105.6 ± 11.9 | 106.5 ± 13.8 | 0.07 [−0.29; 0.43] 0.889 | 96.4 ± 14.6 | 96.6 ± 14.8 | 0.01 [−0.42; 0.44] 0.985 | 9.95 [−2.92; 22.82] | 0.70 [−0.21; 1.61] 0.194 |
| | 120 s | 101.2 ± 15.5 | 104.5 ± 14.3 | 0.22 [−0.12; 0.56] 0.629 | 88.1 ± 11.8 | 94.4 ± 16.6 | 0.44 [−0.55; 1.44] 0.456 | 10.03 [−4.00; 24.05] | 0.65 [−0.26; 1.56] 0.227 |
| 60 *vs* 120 s | ES [CI 90%] *p-value* | 0.32 [−0.09; 0.74] 0.494 | 0.15 [−0.04; 0.33] 0.759 | | 0.63 [−0.31; 1.56] 0.264 | 0.14 [−0.22; 0.49] 0.803 | | | |
| maxLac (mmol/l) | 60 s | 20.7 ± 4.3 | 19.6 ± 3.5 | −0.29 [−0.72; 0.14] 0.531 | 14.4 ± 3.6 | 15.3 ± 3.3 | 0.25 [−0.16; 0.65] 0.667 | 4.29 [1.24; 7.33] | 1.25 [0.28; 2.22] 0.027 |
| | 120 s | 16.0 ± 4.5 | 16.2 ± 3.7 | 0.07 [−0.21; 0.34] 0.886 | 14.0 ± 3.3 | 12.8 ± 3.2 | −0.39 [−0.89; 0.11] 0.526 | 3.48 [0.44; 6.52] | 1.01 [0.07; 1.95] 0.064 |
| 60 *vs* 120 s | ES [CI 90%] *p-value* | 1.08 [0.57; 1.60] 0.018 | 0.93 [0.47; 1.38] 0.068 | | 0.13 [−0.25; 0.50] 0.843 | 0.80 [0.18; 1.38] 0.169 | | | |
| RPE-30 | 60 s | 8.8 ± 1.1 | 8.2 ± 1.1 | −0.51 [−1.16; 0.15] 0.335 | 7.6 ± 1.5 | 7.9 ± 1.2 | 0.21 [−0.87; 1.29] 0.675 | 0.37 [−0.68; 1.41] | 0.32 [−0.58; 1.21] 0.545 |
| | 120 s | 6.7 ± 1.2 | 6.4 ± 1.6 | −0.16 [−0.52; 0.20] 0.739 | 6.1 ± 1.1 | 6.0 ± 1.7 | −0.10 [−0.70; 0.50] 0.859 | 0.44 [−1.05; 1.94] | 0.27 [−0.62; 1.15] 0.607 |
| 60 *vs* 120 s | ES [CI 90%] *p-value* | 1.82 [0.81; 2.84] 0.001 | 1.33 [0.36; 2.29] 0.015 | | 1.10 [0.32; 1.90] 0.038 | 1.26 [0.35; 2.17] 0.041 | | | |
| SmO$_2$T (%) | 60 s | 64.1 ± 6.3 | 61.2 ± 9.6 | −0.36 [−1.05; 0.33] 0.525 | 60.5 ± 11.8 | 42.5 ± 7.0 | −1.92 [−3.05; −0.78] 0.001 | 18.72 [11.42; 26.03] | 2.26 [1.13; 3.39] 0.001 |
| | 120 s | 66.3 ± 10.9 | 61.7 ± 17.2 | −0.33 [−0.73; 0.08] 0.494 | 52.8 ± 7.2 | 50.4 ± 11.6 | −0.26 [−1.22; 0.70] 0.672 | 11.32 [−1.38; 24.01] | 0.79 [−0.13; 1.71] 0.139 |

Benavente et al. (2022), *PeerJ*, DOI 10.7717/peerj.13469

**Table 2** (*continued*)

| | | G1 | | | G2 | | | HH *vs* NH | |
|---|---|---|---|---|---|---|---|---|---|
| | | N | HH | N *vs* HH ES [CI 90%] *p-value* | N | NH | N *vs* NH ES [CI 90%] *p-value* | Adjusted differences between hypoxia types [CI 90%] | ES [CI 90%] *p-value* |
| **60 *vs* 120 s** | **ES [CI 90%]** | **−0.26 [−0.73; 0.21]** | **−0.04 [−0.52; 0.45]** | | **0.81 [0.00; 1.62]** | **−0.85 [−1.86; 0.16]** | | | |
| | *p-value* | *0.616* | *0.943* | | *0.132* | *0.154* | | | |

**Notes.**

60 s/120 s, inter-set rest of the session; N, normoxic condition; HH, hypobaric hypoxia condition; NH, normobaric hypoxia condition; work-HR, heart rate at work; rest-HR, heart rate at rest; HR30, heart rate during the recovery period; maxLac, maximal blood lactate; RPE, rate of perceived exertion; $SmO_2T$, difference between maximal and minimum value of muscle oxygenation during the three sets in total; ES, effect size [calculated as mean difference (H-N or 120-60 s) ÷ (pooled SD) in all dependent variables].

Adjusted between-group difference is the estimated marginal mean of the difference between HH and NH groups (HH group–NH group) after adjusting for N baseline differences.

CI 90%, 90% confidence interval.

Intra- and inter-group ES [CI 90%] are shown in bold.

(adjusted between-group effect of 13.56 bpm (90% CIs [−0.85–27.97 bpm]) and 16.12 bpm (90% CIs [−1.73–33.98 bpm]), respectively for 60 and 120 s inter-set rest intervals).

Maximal blood lactate concentration displayed a moderate decrease as inter-set rest intervals increased in all studied conditions (ES: from 0.6 to 0.9). Compared to HH, NH displayed a moderate to large reduction of the blood lactate accumulation after both types of training sessions (adjusted between-group effect of 4.29 mMol $l^{-1}$ (90% CIs [1.24–7.33 mMol $l^{-1}$]) and 3.48 mMol $l^{-1}$ (90% CIs [0.44–6.52 mMol $l^{-1}$]), respectively for 60 and 120 s inter-set rest intervals).

As expected, ratings of perceived exertion displayed much higher values in 60 s of inter-set rest intervals with respect to 120 s in all conditions (ES: 1.43, 1.33 and 1.26 for N, HH and NH, respectively). There were no differences in the perception of the effort between both modalities of hypoxia.

### Muscle oxygenation

Similar mean $SmO_2T$ values were detected for N and HH at both inter-set rest intervals (ES [p-value]: −0.36 [0.525] and −0.33 [0.494], respectively for 60 and 120 s). NH results displayed a moderate reduction in $SmO_2T$ during 120 s inter-set rest intervals with respect to 60 s (ES = −0.85) (Table 2). Compared to HH, moderate to very large reductions in $SmO_2T$ were observed in NH during both training sessions due to the reduced value in maximal $SmO_2$ reached in the NH group for all sets (adjusted between-group effect of 18.72% (90% CIs [11.42–26.03%]) and 11.32% (90% CIs [−1.38–24.01%]), respectively for 60 and 120 s inter-set rest intervals) (Fig. 2).

## DISCUSSION

The aims of this study were to assess the acute effects of different types of hypoxia (terrestrial *vs* simulated) during a hypertrophy-oriented resistance training session on physiological and performance markers, and to determine whether these responses are affected by alterations in the inter-set rest configuration. As expected, shorter inter-set rest periods increased perceived exertion and produced a moderate increase on cardiovascular and metabolic responses while maintaining muscle performance capacity. Total volume-load for upper- and lower-limbs was similar in both types of hypoxia at each rest condition. For the same inter-set rest configuration, NH considerably decreased the availability of muscle oxygenation among sets and displayed a reduced maximal blood lactate concentration and mean rest-HR compared to HH. These results corroborate previous research (*Millet, Faiss & Pialoux, 2012*) and highlight differences between types of acute hypoxic exposure on the physiological response to $R_T$ exercise (*Millet & Debevec, 2020*). There were no changes in the muscle work capacity among environmental conditions during the $R_T$ session, although the change in the cardio-ventilatory pattern induced by the acute ascent in altitude seems to favor a more immediate recovery in HH compared to NH. Shorter inter-set rest periods produce a more stressful stimulus that, either combined or not combined with hypoxia, affect the acute response to $R_T$ session and conceivably could maximize hypertrophic adaptations in longer periods of training.

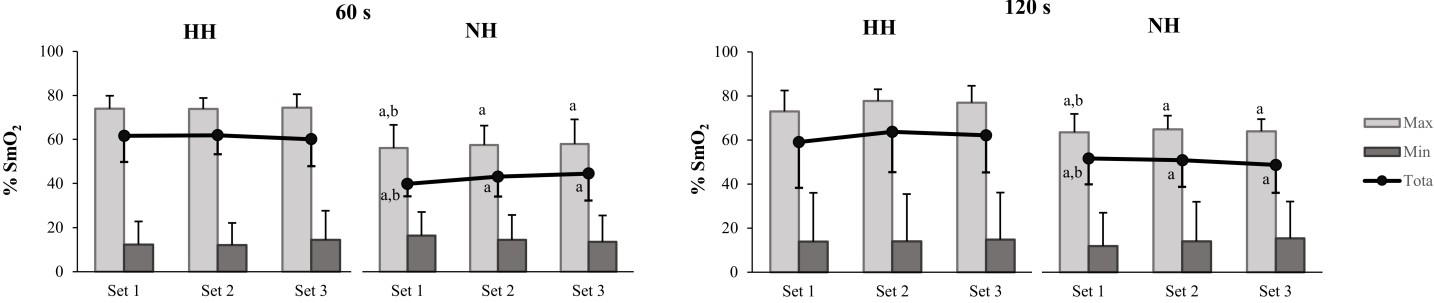

**Figure 2** **Muscle re-oxygenation (max) and de-oxygenation (min) values and the difference between them (total) across the three sets for the barbell back squat.** Mean and SD are represented in both hypoxic conditions (HH and NH) for 60 and 120 s inter-set rest. Significant differences ($p < 0.10$) are displayed between inter-set rest at the same environmental condition ([a]) and between HH and NH ([b]).

Mechanical and metabolic stress are purported influential factors in training-induced development of muscle mass (*Schoenfeld, 2010*; *Schoenfeld, 2013*). Inter-set rest configuration, in combination with volume and intensity, can influence the effectiveness of an acute response or chronic adaptation to a $R_T$ program (*De Salles et al., 2009*). Moderate rest intervals (60–90 s) have been proposed as a viable option for maintaining a balance between mechanical and metabolic factors for gains in strength (*Grgic et al., 2018*) and muscle size (*Grgic et al., 2017*). The present research compares the potential effect of a moderate rest interval (60 s) to a longer rest interval (120 s) during a traditional non-failure $R_T$ program, that preserved mechanical stress between conditions. This outcome was verified by the fact that the total volume-load accumulated during the $R_T$ sessions was quite similar in all environmental conditions and remained almost unaffected by the inter-set recovery periods. The trivial to moderate differences in the total volume-load of the main compared exercises (back squat and bench press) imply a lack of difference in the magnitude of mechanical stress between types of inter-set rest sessions, showing a mean difference ranged from 0.83 to 1.46 kg × set and from 0 to 0.28 repetitions × set in all conditions. Considering that the level of recruitment seemingly cannot provide a mechanistic explanation for the physiological differences between the inter-set rest periods, other factors, such as the observed metabolic effect linked to shorter rest intervals, may be at least partially responsible for these differences (*Wernbom et al., 2007*). Indeed, during the shorter rest intervals, perceptual, metabolic and cardiovascular responses displayed small to large increases across all conditions. As suggested in some studies, it thus is feasible that under relatively equal mechanical load, 60 s-rest intervals provide a more stressful physiological stimulus (*Kraemer et al., 1990*) that potentially could maximize the potential hypertrophic response to $R_T$ under hypoxic conditions. Longitudinal research is needed to test the veracity of this hypothesis.

In contrast to the similarity in the performance between HH and NH in response to both $R_T$ sessions, we observed substantial physiological effects on blood lactate accumulation, rest-HR and $SmO_2T$. Current evidence challenges the traditional assumption that the same inspired partial pressure of $O_2$ produced artificially or by a fall in barometric pressure

produces similar physiological responses (*Richard & Koehle, 2012*; *Millet & Debevec, 2020*). Differences detected between HH and NH suggest an independent barometric pressure effect to the equated partial oxygen pressure, although as noted subsequently, the available acclimatization time to each type of hypoxia condition before exercise could also affect the physiological response.

Throughout the initial hours of exposure to moderate hypoxia there is an increase in ventilation (*Savourey et al., 2003*; *Richard & Koehle, 2012*), submaximal HR and cardiac output (*Hahn & Gore, 2001*). Changes in ventilation induce hypocapnia and develop an alkalotic environment favoring the activation of the glycolytic pathway during exercise. Indeed, the reduction in circulating bicarbonate after a $R_T$ session under hypoxic conditions (*Ramos-Campo et al., 2017*) is interpreted as a higher buffering capacity (*Swenson, 2016*; *Ramos-Campo et al., 2018*). The buffering response may be even more pronounced in HH than in NH due to the differences in the acute hypoxic ventilatory response (*Richard & Koehle, 2012*), which may at least partially help to explain the differences observed in maximal blood lactate between both hypoxic environmental conditions. Ventilatory frequency is known to be greater in HH while the $CO_2$ end-tidal partial pressure is initially lower than in NH (*Savourey et al., 2003*). Preliminary non-published results from our group are in accordance with this finding, showing a 4.98% higher reduction in blood bicarbonate concentration in moderate HH compared to the equivalent NH after a similar $R_T$ session using 60 s of inter-set rest recovery (ES: 0.46; CI [−0.44, 1.36]). Note that the upper limit of the compatibility interval displays a large positive value.

Somewhat counterintuitive, but consistent with some previous research (*Ramos-Campo et al., 2017*; *Scott et al., 2017*; *Feriche et al., 2020*), our results showed a similar maximal blood lactate in N and both types of hypoxia. Blood lactate concentration conceivably should have been higher in H as result of the glycolytic pathway compensation for the reduction in $O_2$ availability in H (*Filopoulos, Cormack & Whyte, 2017*; *Scott et al., 2017*), but remained similar to N due to the slower lactate release from muscle associated with an enhanced buffering response. Otherwise, at NH, maximal lactate concentration displayed a large reduction compared to HH. This decrease could be related to differences in exposure time to the hypoxic stimulus. Consistent with customary practice (*Brocherie et al., 2016*; *Filopoulos, Cormack & Whyte, 2017*), acclimatization to NH only lasted 5 min before the training session. This limited time could constrain adequate activation of the cardio-ventilatory compensation mechanisms and, therefore, of the buffering response, limiting the hypoxic effect on maximal lactate accumulation. The large lower $SpO_2$ reached at moderate NH compared to HH just before the start of the training session is consistent with this approach ($SpO_2$: 94.3 and 90.7%, respectively for HH and NH, ES = −3.29, $p = 0.001$) displaying differences in the severity of internal hypoxia achieved in each group for the same external hypoxia ($FiO_2$ of 15.9%) (*Soo et al., 2020*). The short connection time to the hypoxic system in NH before the start of the training sessions (most frequent connection times are ranged between 5 and 10 min) could cause a greater work of breathing in the participants due to the abrupt increase in flow rates and the higher gas density, reducing the acclimatization of the ventilatory response in comparison to the HH group (*Richard & Koehle, 2012*). After longer exposures (~1 h), and according to the

SpO$_2$ observed at the end of the 30 min of recovery (SpO$_2$: 91.7 and 94.5%, respectively for HH and NH, ES = 1.35, $p = 0.002$), desaturation is usually greater in HH (*Savourey et al., 2003*).

To our knowledge, there currently are no data in the literature on the impact of the type of hypoxia (terrestrial *vs* simulated) on muscle oxygenation. Compared to normoxia, severe NH (FiO$_2$ = 13%) reduces muscle oxygenation from the vastus lateralis when performing the leg press (5x10 rep; 70% 1RM; 60 s rest) (*Kon et al., 2010*) and from the triceps brachii after performing shoulder press and bench press (3–6 × 10 rep; ~75% 1RM; 60 s rest) (*Walden et al., 2020*). Contrarily, similar mean relative values from the vastus lateralis oxygenation between moderate NH (FiO$_2$ = 15–16%) and N have been observed in other studies (*Scott et al., 2017*; *Lockhart et al., 2020*) after 3–5 sets × 10 repetitions (60–70% 1RM; 60 to 180 s rest) of lower-limb exercises (leg press, back squat or deadlift). These discrepancies among studies could be due to differences in the muscle assessed, type and/or severity of hypoxia when the training session is performed at simulated hypoxia. In our results, the minimum, maximum and total SmO$_2$ changes from the vastus lateralis were not affected by the inter-set rest duration at any environmental condition.

Compared to HH and N, moderate to very large reductions in the muscular maximal reoxygenation response during the back squat exercise were observed in NH. Surprisingly, the muscle reoxygenation capacity during the HH sets was similar to N. The accentuated increase in cardiac output and buffering capacity described in acute terrestrial hypoxia, compared to simulated hypoxia, is likely to improve glycolytic ATP production and promote muscle perfusion during recovery (*Kawada, 2005*; *Richard & Koehle, 2012*; *Feriche et al., 2020*). Moreover, the oxygen release in active muscles is favored by a rightward shift of the oxyhemoglobin curve (Bohr Effect) during exercise in H (*Gerbino, Ward & Whipp, 1996*), which can also enhance the reoxygenation of muscle tissue at HH due to the large reduction in pH after exercise (*Richard & Koehle, 2012*). Research suggests 15–16% of FiO$_2$ as the minimum threshold for inducing changes in the muscle oxygenation (*Lockhart et al., 2020*; *Walden et al., 2020*). Our results in NH do not support this hypothesis, although future studies are necessary to clarify the influence of the severity, type and time of exposure to hypoxia on muscle oxygenation in a similar R$_T$ session configuration.

This study has some potential limitations: (1) A double-blind design could not be employed in the HH group due to the intrinsic characteristic of the terrestrial altitude. To reduce the potential for confounding, participants were not informed about the expected hypoxic effect on performance. (2) Blood lactate concentration itself may provide a limited understanding of the magnitude of exercise-induced metabolic stress due to the dissociation between the intra and extra muscular response (*Lockhart et al., 2020*), as well as the fact that lactate represents just one of dozens of metabolites produced during exercise (*Schranner et al., 2020*). (3) Variation in vastus lateralis oxygenation was only assessed in this muscle during the first exercise. The analysis of other upper-limb muscles during a full-body traditional hypertrophy session, such as the used in this study, could provide additional information of interest on this variable. (4) Our sample size was relatively low, which could have influenced the width of probability distributions across outcomes. However, this population is understudied and of specific interest, and the research-based methods and

commitment for data collection are encumbering for these individuals. (5) Although all the participants performed the same $R_T$ program and the total volume load accumulated was similar between conditions, it is possible that one condition was closer to momentary muscle failure than the other in some exercises.

## CONCLUSIONS

In conclusion, shorter session's inter-set rest intervals (60 s) provide a more potent cardiovascular and metabolic stimulus and intensify the perceptual response in all environmental conditions. For an equivalent $FiO_2$, the type of hypoxia (terrestrial *vs* simulated) affects the physiological response to a traditional hypertrophy-oriented $R_T$ session. The improvement in buffering capacity and rest-HR at HH favors a better inter-set recovery compared to NH, with findings more prominent as the rest intervals shorten. In addition, it is possible that the 5 min of pre-exercise acclimatization time provided in NH constrained the activation of the physiological compensation mechanisms affecting the muscle oxygen saturation.

Although our results provide intriguing insights into the physiological response to rest periods under different types of hypoxias, the acute design precludes the ability to extrapolate findings to long-term adaptations at the studied conditions. A different stressful response to the same exercise conceivably could occur with a longer acclimatization time. Future research should aim to determine whether more severe simulated hypoxia or longer pre-exercise exposure times are required in NH to promote equal physiological responses and muscle adaptations to HH during resistance training.

## ACKNOWLEDGEMENTS

The authors thank the High-Performance Center of Sierra Nevada, Spain and all the participants who volunteered for this investigation. The authors also thank Dymatize Europe and Vithas Granada for respectively supplying the meal replacement and blood collection equipment in this study.

### Funding
This work was supported by the Spanish Ministry of Science, Innovation and Universities under grant (PGC2018-097388-B-I00-MCI/AEI/FEDER,UE), by the Andalusian FEDER Operational Program (A-SEJ-246-UGR18 & B-CTS-374-UGR20) and FPU pre-doctoral grant (FPU18/00686) awarded to one of the authors. The funders had no role in study design, data collection and analysis, decision to publish, or preparation of the manuscript.

### Grant Disclosures
The following grant information was disclosed by the authors:
Spanish Ministry of Science, Innovation and Universities: PGC2018-097388-B-I00-MCI/AEI/FEDER,UE.

Andalusian FEDER Operational Program: A-SEJ-246-UGR18 & B-CTS-374-UGR20. FPU pre-doctoral: FPU18/00686.

## Competing Interests

Brad J. Schoenfeld serves on the scientific advisory board to Tonal Corporation, a manufacturer of exercise equipment.

## Author Contributions

- Cristina Benavente performed the experiments, analyzed the data, prepared figures and/or tables, authored or reviewed drafts of the paper, and approved the final draft.
- Belén Feriche conceived and designed the experiments, performed the experiments, analyzed the data, prepared figures and/or tables, authored or reviewed drafts of the paper, and approved the final draft.
- Guillermo Olcina conceived and designed the experiments, performed the experiments, authored or reviewed drafts of the paper, and approved the final draft.
- Brad J. Schoenfeld analyzed the data, prepared figures and/or tables, authored or reviewed drafts of the paper, and approved the final draft.
- Alba Camacho-Cardenosa performed the experiments, authored or reviewed drafts of the paper, and approved the final draft.
- Filipa Almeida performed the experiments, analyzed the data, prepared figures and/or tables, authored or reviewed drafts of the paper, and approved the final draft.
- Ismael Martínez-Guardado performed the experiments, authored or reviewed drafts of the paper, and approved the final draft.
- Rafael Timon conceived and designed the experiments, performed the experiments, authored or reviewed drafts of the paper, and approved the final draft.
- Paulino Padial conceived and designed the experiments, performed the experiments, prepared figures and/or tables, authored or reviewed drafts of the paper, and approved the final draft.

## Human Ethics

The following information was supplied relating to ethical approvals (i.e., approving body and any reference numbers):

This study was approved by the Andalusian Government Research Ethics Committee and conducted in accordance with the Helsinki Declaration.

## Data Availability

The raw measurements are available in the Supplementary File.

## Supplemental Information

Supplemental information for this article can be found online at http://dx.doi.org/10.7717/peerj.13469#supplemental-information.

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
