# Peer review of "Inter-set rest configuration effect on acute physiological and performance-related responses to a resistance training session in terrestrial vs simulated hypoxia"

_PeerJ, doi:10.7717/peerj.13469_

## Round 0.1 · original submission · Major Revisions

The reviewers found your manuscript of interest but all 3 have provided recommendations to improve the paper (e.g. with some of the methodology ).

Reviewer 1 ·

Basic reporting

This paper is generally well-written. However, there are a few sections where methods are not clearly explained (highlighted below). Please amend these sections to improve the clarity of this paper.

Experimental design

Congratulations on an interesting study, and on the comprehensive assessments completed. However, I have a few queries on the research design:

Did the authors conduct an a priori sample size calculation? It is possible that this study is limited by statistical power with between group comparisons, as these groups were relatively small (G1=9 and G2=7). Also, this may have accounted for some differences in the responses to NH vs HH (i.e. between-participant differences in how they responded to NH or HH)

I am a little confused with some parts of the manuscript that indicate either two or three barbell exercises were analysed only, when there were six exercises performed during exercise. More clarity is required in the explanations of this throughout.

The emphasis on the resistance training volume seems a little strange to me, considering that participants were instructed to perform 3 sets of 10 reps. If all sets were performed to failure I would understand the logic in this, but it seems that the volume of exercise was somewhat fixed. This may also mean that discussion section from lines 298-310 needs to be amended.

Validity of the findings

The conclusions are in accordance with the data. In my opinion, the discussion is well constructed and raises some interesting points (i.e. the rationale for differences between HH and NH are well explained).

Additional comments

• Line 68-69: the part of this sentence “…although other factors may be involved as well…” is a little vague. Can the authors make this statement more specific to what these factors are?
• Line 85-86: The term “…being unknown the effects…” is a little confusing. Please re-word this to improve clarity.
• Line 89-91: The study from Lockhart et al. (which has been cited by the authors at other parts of this paper) compared long vs short rest periods in normobaric hypoxia vs normoxia on acute physiological responses which are relevant to this study. It may be worth mentioning that study here as the only previous one which has examined the effects of different rest periods on these responses during hypoxic resistance exercise.
• Line 95: the grammar of this sentence should be improved.
• Line 98: Probably need to be careful referring to “the hypertrophic response” – this study examined acute physiological responses which may be related to muscle growth, but it did not assess the hypertrophic response, per se.
• Line 103: what is meant by the term “acute muscle performance response”? perhaps this can be re-worded to improve the clarity of this statement.
• Line 118-119: the section on “the mean of two attempts separated by 15 mins” is a little confusing – was there only 2 attempts to lift 1RM with 15 min rest between them?
• Line 127: the term “training” is typically used to refer to repeated exercise bouts culminating in an adaptation (i.e. a 6-week training program), whereas “exercise” may be a more appropriate term to describe an acute session (i.e. exercise sessions were conducted….”). Please amend this throughout the manuscript so that the term “training” is not sed to refer to a single acute exercise bout.
• Line 129-131: I am a little confused about what is meant by the “hypoxic environmental requirement” – Please re-word this to make it clearer.
• Line 158: please amend “…the load was reduced a 5% as…” to improve grammar.
• Line 160: I assume these three exercises were the squat, bench press and military press? why were just these three used for between-condition comparisons?
• Line 163: were participants instructed to lift to failure, or to perform sets of 10 repetitions (as described on line 155)?
• Line 172: It may be clearer that the HH and NH conditions were somewhat matched if you also include that the high performance facility is 2320 m asl. You could alternatively mention this in line 177.
• Line 189: may be more correct to call this a “sessional rating of perceived exertion”.
• Line 198: here the sampling frequency is indicated as 2Hz, but line 191 suggests this at 1Hz (i.e. second-by-second). Please clarify and amend. Also, it is not clear what you mean by “internal memory values”? I assume this is just simply the SmO2 %?
• Line 236: this refers to 2 barbell exercises, whereas line 0160 refers to three exercises?
• Line 237: please clarify how the ADJUSTED between group effects were calculated? I found this in the description for Table 1, but it should probably be within the statistical analysis section of the paper.
• Line 396: another limitation which should be addressed is the between group comparisons of HH vs NH. It is possible that this, along with the relatively small sample size in each group (9 and 7) introduced additional variability into the data.

Reviewer 2 ·

Basic reporting

This is an interesting experiment to examine the possible influences of two different types of hypoxic environments and two different durations of interest resting on heart rate, blood lactate, muscle oxygenation, and delayed RPE. I applaud the authors for their review of the literature in the introduction and discussion. Additionally, the manuscript is written in a professional, unambiguous language. However, there are some organizations to the ideas that may be better portrayed if subsections or gatherings of similar ideas are grouped in each paragraph.

Line 56: The first paragraph provided a clear brief on setting up the interest of the study but then there is a conclusion of the paragraph where the rest intervals are important because of the role in the phenotypic response of resistance training. This does not seem to follow smoothly into the next paragraph, and it would seem almost like an added thought. Since phenotypes are not the interest in this study, I would recommend adding a conclusive sentence to this paragraph to why the authors find the testing of rest intervals is important in regard to what the authors plan to test.

Lines 65 to 69: The information in this sentence is important to understand the rationale to why specific variables are tested; the sentence should be shortened to where “although other factors may be involved as well…” should be a stand-alone sentence.

Line 129–131: It would be helpful to have a short expansion on what the “requirement” for the hypoxic environment was.

Line 158: “The load was reduced [by] 5 % as …”

Line 220: “90% compatibility intervals” is abbreviated as (CIs). This is awfully close to the abbreviation, CI, that is used as confidence intervals. Since “CIs” is used about 3 times in the manuscript, it may be better to type it out instead of abbreviating it.

Lines 249–264: I find that it may be a more organized way to read if this was separated into three separate subsections.

Line 396: It may be of interest to include a brief mention of the small sample size

Experimental design

Lines 95–96: In the indication of what the current study will provide in the literature, it should include the term “hypoxia” since this study is specific to that. Additionally, this study is an acute study and the change in muscle mass and strength is not being examined here. Thus, the results of this study would not be able to determine training variables that are suitable for maximal muscle mass gains and strength gains. It would be of consideration that the rationale for how this study contributes to the literature should be restated.

Line 189–190: It would be of interest to add in why the RPE was taken 30 minutes after the exercise. Generally, the RPE is how a participant perceives his/her exertion level at the moment. Therefore, 30 minutes after would no longer seem to be the normal use of RPE. Although Foster et al. 2001 is cited, it would be best to provide some description of why this was used and how this would be the best method in answering your current questions. For example, this could be called a “delayed RPE” and I believe the explanation that was used in Foster was that it was meant to not have participants who would not drastically rate based on difficult or easy segments in the exercise bout.

Validity of the findings

Line 167–168: Excuse me as I am a little perplexed in reading the average SpO2 separated into 3 different groups. Since the “N” group consisted of 9 individuals from “HH” and 7 individuals from “NH” totaling to 16 individuals in the “N” group and then separate means for the “HH” and the “NH” group which would seemingly indicate that there was (16 + 9 +7) 32 individuals that were accounted for the sample size. Since there was mentioning of G1 and G2, then the average here should be expressed as so.

In the results section: I see the rationale with reporting the effect sizes while providing indications of how meaningful the findings were but that does not allow the reader to know what changes have occurred. As a suggestion, it would be of interest to add in the average changes and SD. Thus, the reader can see what the values were for the dependent variables and how meaningful it is via the ES.

Lines 347–348: What makes “blood lactate concentration […] should have been higher in N”? This may be better if expanded in order to explain the idea and then continue to explain why the researchers have observed, otherwise.

Line 408: In the sentence “[…] shorter session’s inter-set rest intervals (60 s) provide a more potent perceptual”, what is potent perceptual? Is it more so of a potent perception?

Lines 409–410: It may be difficult to state that this “could maximize hypertrophic adaptations in longer periods of training” because this was specifically an acute test and the conclusions may be better if it stays within what was tested. This is mentioned further along with the conclusion so it may be best to not state this at all.

Annotated reviews are not available for download in order to protect the identity of reviewers who chose to remain anonymous.

Reviewer 3 ·

Basic reporting

I commend the authors on their work. This is an interesting and overall well written manuscript. I found many aspects of this manuscript interesting and I hope the authors find my feedback helpful.

Introduction:

1. Introduction Page 6 lines 54-59 “Training volume and load are considered primary factors to maximize strength and hypertrophy (Kraemer and Ratamess 2004), but other variables, such as rest intervals, also play a role in the phenotypic response to RT.”– I am not sure that training volume is an important factor for strength. Further, it does not seem like load is an important factor for hypertrophy. Rather, it seems that specificity of load is important for strength and a sufficient amount of activation/recruitment is necessary for growth. I would modify this slightly.

2. Intro line 59-62 – “Evidence suggests that RT performed under hypoxic conditions may produce an added benefit to strength and muscle mass development compared to an equivalent amount of training under normoxic conditions….This benefit is purportedly linked to the augmented accumulation of metabolic byproducts at hypoxia, such as blood lactate or protons (H+) among others, derived from the increase in anaerobic metabolism to compensate the loss of oxygen (O2) availability” – Are these benefits similar for strength and hypertrophy? I believe that for strength the data is a little less consistent. In addition, authors cite work of Manimmanakorn; however, there is other research that should be considered that does not find a difference between BFR conditions and non-BFR conditions when exercise is performed to failure. It may be more appropriate to cite research that performed the same number of reps and compared BFR or hypoxic conditions to non-BFR hypoxic conditions (considering that authors state when an equivalent amount of training is performed).

3. There is a lot of discussion of metabolic contributions in the introduction, which led me to suspect that BFR was going to be included in this study. I think it is important to consider that pooling metabolites within a muscle through limiting venous return with BFR is likely different compared to mechanisms associated with hypoxia. Some clarification on the metabolite response to hypoxia and the ability to clear the metabolites during rest periods under hypoxia would be beneficial

4. Introduction, Page 7 lines 82-83: “Contrarily, inter-set rest periods longer than 90 s seem to limit the potential hypoxic benefit on the muscle response after a single RT session (Scott et al. 2015a, b)”….” I think this statement may need a little elaboration as it seems important regarding the rationale of this paper. However, the Scott 2015b does not seem to support this statement as the authors only examined 3 minutes of rest. Their review paper seems to flush out the topic and I think that some of the rational behind this idea can be presented a little better in the introduction. The Kon and Ho papers from 2014 also do not compare rest periods. Thus, perhaps clarify that some studies utilizing short rest periods have observed differences whereas, studies utilizing longer rest periods have not observed differences.

5. Regarding comment 4 (above) do all of the studies match repetitions or have individuals train to failure? This would be an important consideration when discussing these studies. For comparison, sometimes BFR appears beneficial if 2 groups are lifting the same load and completing the same repetitions. However, if repetitions are performed until failure, these differences largely seem to disappear.

Experimental design

6. Methods – Authors state that “Absolute training load by exercise (kg) and repetitions to failure were monitored during each training session.” However, it is previously stated that 3sets of 10 repetitions at 70%1RM are performed on each exercise. Can the authors please clarify?

7. Following up on the previous comment, it seems like some individuals were training to failure and others weren’t. Can the authors report what percent of sets in each training condition were performed to failure?

8. Perceptual Responses – Why was perceptual response taken 30 minutes following the exercise bout and not during the exercise bouts? Can Authors please clarify?

Validity of the findings

Discussion/Analysis/Conclusions
9. “The present research compares the potential effect of a moderate rest interval (60 s) to a longer rest interval (120 s) during a traditional non-failure RT program, that preserved mechanical stress between conditions.” - I am unsure of this comment unless it is clarified how often individuals were actually training to failure.

10. Can the authors provide clarification on the statistical analysis? My understanding is that using the pre-test standard deviation or pooled pre-test and post-test standard deviation for a specific variable (although often used in exercise science) does not explain the variability of the training effect, but rather the variability of the study population. Instead it has been suggested that the between group Cohen’s d may be more appropriate. It seems that the beetween group Cohen’s d was sometimes (I think only for training volume)….but not always reported. https://pubmed.ncbi.nlm.nih.gov/28277241/


11. Discussion: Page 13 Lines 311-314 – “Considering that the level of recruitment seemingly cannot provide a mechanistic explanation for the physiological differences between the inter-set rest periods, other factors, such as the observed metabolic effect linked to shorter rest intervals, may be at least partially responsible for these differences” – I am not sure this is necessarily true as the proximity to failure may be different between the 2 conditions. Might this infer differences in recruitment? If one condition was training to failure and the other wasn’t than metabolites potentially had an indirect effect on recruitment or overall stress

12. The fact that some sets were taken to failure and others weren’t may be perceived as a limitation. This difference in training conditions may dilute differences caused by the intervention.

13. Conclusion “In conclusion, shorter session’s inter-set rest intervals (60 s) provide a more potent perceptual, cardiovascular and metabolic stimulus in all environmental conditions, which could maximize hypertrophic adaptations in longer periods of training. – I am not sure that this statement is supported in this study. Authors do not discuss differences in hypertrophic potential in the discussion, so I was surprised to see this in the conclusion. I think this statement should be tempered a bit. It is acknowledged later in the conclusion with more caution, but may not belong in the first sentence.

---

## Round 0.2 · Minor Revisions

The reviewers have suggested a few additional points that should be considered.

Reviewer 1 ·

Basic reporting

The authors have done a nice job of amending their paper and improving the clarity based on feedback. Thank you for making the suggested changes.

Congratulations again on an interesting study.

Experimental design

No further comments

Validity of the findings

No further comments

Reviewer 2 ·

Basic reporting

The authors of this manuscript have provided an additional clear and professional structure of the material in the introduction, methods, results, and discussion. The additional explanations to the specific concerns as mentioned prior were amended and adds a greater understanding of the reason specific methods were implemented. Overall, excellent revision.

Experimental design

The research question is well-defined and meaningful. This allows a specific population, already strength-trained individuals, to be evaluated. The methods added provide sufficient detail and information.

Lines 464 to 466: I think it is great to mention that the sample size was small but please add a sentence after it to provide justification. The authors did a great job providing an explanation in the rebuttal and this should be briefly added. Do not sell yourself short. Something along the lines of "However, this specific population is of interest, and the current methods and commitment are encumbering for this population". Although this was not done, as a side note, it may not be of good consideration to add on how other research did this, so we did as well.

Validity of the findings

The conclusions were well stated and linked to the research question.

Reviewer 3 ·

Basic reporting

The authors have addressed the majority of my concerns. I have a few additional comments that I hope the authors find constructive.

1. Authors may consider revising the wording of their hypothesis statement. The use of the phrasing “greater physiological stimulus” is a little vague. For example, authors may consider saying “We hypothesized that short rest periods would produce greater changes in blood lactate, greater percetual responses and…..”.

2. Page 6, Line 216- The word government is spelled incorrectly as goverment

Experimental design

No Comment

Validity of the findings

3. Statistical Analysis: Please clarify which SD was used for calculation of effect sizes. Was it the pre SD or the SD of the change?

Additional comments

4. Discussion Page 11, Lines 417-419: Although the total volume load accumulated was similar, isn’t it possible that one condition was closer to failure than the other? For example, if one condition completed all repetitions and could have (if allowed) performed an additional 5 reps on each set than this condition was potentially underdosed. The authors seem fixated on volume, which was similar (in part) by the design. I think that this should at least be acknowledged in the limitation section of the manuscript.

“This outcome was verified by the fact that the total volume-load accumulated during the RT sessions was quite similar in all environmental conditions and remained almost unaffected by the inter-set recovery periods.”

---

## Round 0.3 · Minor Revisions

There was one very minor point the reviewer raised that I too believe could be further clarified.

Reviewer 3 ·

Basic reporting

Authors have addressed my concerns. I have only one final minor comment.

Introduction: The sentence below seems to be missing and ending. Do authors mean that strong inferences cannot be drawn with regards to more complex or metabolically demanding protocols?
“The only previous study to examine the effects of different rest periods during RT under H conditions (Lockhart et al. 2020) used a single-joint exercise, thus limiting the ability to draw strong inferences as to whether short rest intervals combined with different types of hypoxia may enhance muscular adaptations.”

Experimental design

No comment

Validity of the findings

No comment

Additional comments

No Comment

---

## Round 0.4 · accepted · Accept

Thank you for addressing the concern of the reviewer. The sentence you added is difficult to follow but I think this can be edited in the proof stage.